# Effects of Forest Fragmentation on the Vertical Stratification of Neotropical Bats

**Inês Silva** [1,2,3,*,†] , **Ricardo Rocha** [2,3,4,†] , **Adrià López-Baucells** [2,3,5] , **Fábio Z. Farneda** [2,3,6]
**and Christoph F. J. Meyer** [2,3,7]

1    Conservation Ecology Program, School of Bioresources and Technology, King Mongkut's University of Technology Thonburi, Bangkhunthien, Bangkok 10150, Thailand

2    Centre for Ecology, Evolution and Environmental Changes – cE3c, Faculty of Sciences, University of Lisbon, 1749-016 Lisbon, Portugal; ricardo.nature@gmail.com (R.R.); adria.baucells@gmail.com (A.L.-B.); fabiozfarneda@gmail.com (F.Z.F.); C.F.J.Meyer@salford.ac.uk (C.F.J.M.)

3    Biological Dynamics of Forest Fragments Project, National Institute for Amazonian Research and Smithsonian Tropical Research Institute, Manaus 69011-970, Brazil

4    CIBIO/InBIO-UP, Research Centre in Biodiversity and Genetic Resources, University of Porto, 4485-661 Vairão, Portugal

5    Granollers Museum of Natural Sciences, Granollers 08402, Spain

6    Department of Ecology, Federal University of Rio de Janeiro, Rio de Janeiro 21941-902, Brazil

7    School of Science, Engineering and Environment, University of Salford, M5 4WT Salford, UK

\*    Correspondence: imss.silva@gmail.com

†    These authors have contributed equally to this work.

**Abstract:** Vertical stratification is a key component of the biological complexity of rainforests. Understanding community- and species-level responses to disturbance across forest strata is paramount for evidence-based conservation and management. However, even for bats, known to extensively explore multiple layers of the complex three-dimensional forest space, studies are biased towards understory-based surveys and only few assessments of vertical stratification were done in fragmented landscapes. Using both ground and canopy mist-nets, we investigated how the vertical structure of bat assemblages is influenced by forest fragmentation in the experimentally fragmented landscape of the Biological Dynamics of Forest Fragments Project, Central Amazon, Brazil. Over a three year-period, we captured 3077 individuals of 46 species in continuous forest (CF) and in 1, 10 and 100 ha forest fragments. In both CF and forest fragments, the upper forest strata sustained more diverse bat assemblages than the equivalent understory layer, and the midstory layers had significantly higher bat abundance in fragments than in CF. *Artibeus lituratus* and *Rhinophylla pumilio* exhibited significant shifts in their vertical stratification patterns between CF and fragments (e.g., *R. pumilio* was more associated with the upper strata in fragments than in CF). Altogether, our study suggests that fragmentation modulates the vertical stratification of bat assemblages.

**Keywords:** Amazon; Chiroptera; community ecology; deforestation; Neotropics; vertical space

## 1. Introduction

Tropical forests harbor ca. 60% of all known animal and plant species in only 8% of the planet's surface [1]. This diversity is largely mediated by the complex stratification and multidimensionality of tropical forest canopies, which allow for additional niche space and facilitate the coexistence of a large number of species in the same geographical area [2–4]. However, biological assessments across the tropics tend to be largely limited to understory-level surveys that under-represent species associated with higher forest strata [5]. While these have provided important insights into the responses

of tropical rainforest vertebrates to disturbance (e.g., [6]), they likely underestimate diversity and abundance levels and thus give an incomplete picture of the responses of rainforest communities to forest degradation [5,7].

The Amazon basin holds ca. 40% of the planet's remaining tropical forest and is home to a disproportionate amount of biological diversity [7]. Yet, due to multiple and often interacting stressors, the region is facing rapid environmental change and since 1970 has lost over 790,000 km$^2$ (nearly 20%) of its original forest cover [8]. Although habitat loss and fragmentation continue to act as primary threats to the megadiverse Amazonian vertebrate communities [9–11], little is known about how their composition and structure across forest strata is affected by habitat modification.

With roughly 1400 species globally [12], bats are the second largest mammalian order and account for 25% of the total mammal diversity in the Brazilian Amazon [13,14]. Powered flight allows bats to explore resources across the multilayered space of tropical rainforests and an increasing number of studies have documented changes in species abundance from ground to subcanopy and canopy levels in both the Neo- and Paleotropics (e.g., [15–18]). Bats provide vital ecosystem services as seed dispersers, pollinators and arthropod predators [19] and, given the strong preference of several frugivorous species for pioneer plants, promote the regeneration of disturbed areas [20,21]. They are acutely sensitive to human-induced forest disturbances [22] and have extensively been used as an indicator group for evaluating the effects of habitat fragmentation on tropical biota [23,24].

The responses of tropical bat assemblages to forest fragmentation are to a large extent species-, ensemble- and habitat-specific (reviewed in [23]). For example, gleaning animalivorous bats are regarded as more susceptible to fragmentation and disturbance than either frugivores or nectarivores [25,26]. However, despite known effects of fragmentation on the vertical stratification of forest invertebrates (e.g., [27]), we know little about the effects of fragmentation on the vertical stratification of bat assemblages (but see [28,29]) as most bat vertical stratification studies targeted only continuous forest [16,30–35] or forest fragments [36]. Yet, these studies have shown that species richness and abundance differ among strata, and that some species can be classified as either understory or canopy specialists. Canopy foragers appear to be less sensitive to fragmentation than understory species, as they tend to be more mobile due to less evenly distributed resources [29,34,37].

Here, we combined extensive ground and canopy mist-netting to explore the effects of forest fragmentation on the vertical stratification of bat assemblages within a landscape-wide fragmentation experiment in the Brazilian Amazon. Specifically, we addressed the following questions:

1. How does bat diversity, abundance and assemblage composition change between the understory and upper strata of continuous forest (CF) relative to different-sized (1, 10 and 100 ha) forest fragments? We predicted higher diversity across strata in CF and 100 ha fragments than in the small (10 and 1 ha) fragments, and across upper forest strata, relative to the understory. Additionally, we anticipated higher turnover of species within fragments and lower forest strata than in CF and upper forest strata.
2. Which species are more often captured in the upper forest strata in relation to the understory? We expected to have higher capture rates of the Stenodermatinae subfamily in the upper forest strata, due to their preference for fruit tree species present in the subcanopy.
3. How do stratification and fragmentation interact as predictors of both species richness and abundance? We hypothesized that there is a combined effect of stratification and fragmentation for certain ensembles (i.e., gleaning animalivores, frugivores), given species-specific associations with certain strata.

## 2. Materials and Methods

### 2.1. Study Area

The study was conducted at the Biological Dynamics of Forest Fragments Project (BDFFP), a 1000 km$^2$ reserve located 80 km north of Manaus, state of Amazonas, Brazil (2°24′ S, 59° W;

Figure S1). Established in 1979, the BDFFP is considered the world's largest and longest-running experimental study on the impact of forest fragmentation on tropical biota [7]. The 80-650 m that separate the experimental forest fragments from CF have already been shown to induce multiple fragmentation-driven changes to the habitat structure of forest fragments and to the composition and abundance of both phyllostomid [26] and non-phyllostomid bats [29]. Canopy is 30–37 m tall, with emergent trees reaching 55 m. Local climate corresponds to Köppen's Af type, with an average annual temperature of 27 °C (maximum: 35–39 °C, minimum: 19–21 °C), and a well-defined dry season from June to October when precipitation drops below 100 mm/month and a rainy season from November to May when precipitation can exceed 300 mm/month [38].

Bats were sampled in 17 sites: nine CF sites in three areas of continuous lowland *terra firme* rainforest (Cabo Frio, Florestal and Km 41 camps), and the interiors of eight forest fragments (three 1 ha, three 10 ha, and two 100 ha; Colosso, Porto Alegre and Dimona camps). All fragments were initially isolated from nearby intact forest in the early 1980s and are now surrounded by a matrix of secondary regrowth forest [39]. To maintain isolation, a 100 m-wide area around each fragment was cleared on 3–4 occasions prior this study (most recently between 1999 and 2001) and again in 2014 [40].

*2.2. Data Collection*

We visited each sampling site between eight and 12 times between August 2011 and November 2014, for a total of 191 sampling nights, and captured bats using 14 mist-nets set at ground level (12 × 3 m) and two to three mist-nets set at subcanopy level (2.5 × 12 m; average maximum height in CF sites (mean ± SE): 17.88 ± 0.25 m, and fragmented sites: 17.35 ± 0.22 m). We opened mist nets each night between dusk (~1800h) and midnight, and inspected them every 15 to 30 min. Species identification and nomenclature follow López-Baucells et al. [14], except for *Pteronotus* cf. *parnellii*, *Lonchophylla thomasi* and *Mimon crenulatum* which, based on recent taxonomic work, are referred to as *Pteronotus* cf. *rubiginosus* (*sensu* [41]), *Hsunycteris thomasi* and *Gardnerycteris crenulatum*, respectively.

We classified all species into the following ensembles: gleaning animalivores, frugivores, nectarivores, sanguivores, and aerial insectivores [26]. Additionally, we assigned captures to four strata: understory (U; < 3 m), lower midstory (LM; 3–9 m), upper midstory (UM; 9–15 m) and subcanopy (C; >15 m). Since most mist-net surveys tend to be restricted to the understory layer (< 3 m), for some of the analyses we contrast the understory with pooled data of both midstory layers and subcanopy (i.e., "Upper Strata (all)"). As mist-netting in the Neotropics is only an effective sampling method for phyllostomid and mormoopid bats [42], all analyses were restricted to phyllostomid species and *Pteronotus* cf. *rubiginosus*. Bat capture and handling was conducted following guidelines approved by the American Society of Mammalogists [43] and in accordance with Brazilian conservation and animal welfare laws

*2.3. Data Analyses*

To assess inventory completeness, we calculated randomized (1,000 iterations) sample-based rarefaction curves using EstimateS software version 9.1 [44], and the non-parametric richness estimator Jackknife 1, due to its low-bias estimation even at small sample sizes [45]. Jackknife 1 also considers the movement heterogeneity of highly mobile animals such as bats [46], and performed well in comparisons with other estimators in a similar phyllostomid bat assemblage study [32].

2.3.1. Species Richness, Diversity and Dominance

As measures of diversity, we used Hill numbers, or the effective number of species (*q*; [47–49]). Specifically, we calculated the first three Hill numbers: species richness (*q* = 0; insensitive to species frequencies), the exponential of Shannon's entropy index or Shannon diversity (*q* = 1; weighting species in proportion to their frequency), and the inverse of Simpson's diversity (*q* = 2; placing greater weight on the frequencies of dominant species). We evaluated statistical differences in these diversity metrics between stratum (understory, lower midstory, upper midstory, subcanopy) and habitat categories (i.e.,

CF, 1 ha, 10 ha, 100 ha), based on the 95% confidence intervals derived using the package 'iNEXT' in R [50].

We employed generalized linear mixed models (GLMMs, [51]) using the 'lme4' package in R [52] to test for differences in species richness between stratum and among habitat categories. We used a Gaussian error distribution (log link) for species richness (as Hill number $q = 0$; obtained through iNEXT). Sampling effort [1 mist-net hour (mnh) equals one 12-m net open for one hour] was included as an offset to account for differences in sampling effort. Habitat category (i.e., CF, 1 ha, 10 ha, 100 ha) and stratum (i.e., understory, lower midstory, upper midstory, subcanopy) were specified as fixed effects, and modeled as single-variable, additive and interactive models. We further incorporated location as a random effect (i.e., sampling sites nested within the different camps; Figure S1) to account for potential spatial autocorrelation [53]. We also performed additional GLMMs at the ensemble level, with species richness within each guild as the response variable (family gaussian with log link); however, both sanguivores and aerial insectivores were not considered as they comprised only one species each.

### 2.3.2. Species Composition and Abundance

We calculated rank-abundance curves and performed pairwise comparisons for each stratum between CF and 1 ha, 10 ha, and 100 ha fragment sizes using Anderson-Darling k-sample tests with a Bonferroni correction, using the 'kSamples' R package [54]. To further assess compositional differences between CF and fragments across the four forest strata, we calculated species turnover and mean rank shifts as two measures of community dynamics, using the package 'codyn' in R [55], whereby habitat category was specified as our "temporal" variable. Turnover is here defined as the rate at which species appear and disappear between CF and successively smaller fragments [56], while mean rank shift is defined as relative changes in species rank abundances [57].

We ran another set of GLMMs to test for differences in total abundance between strata and among habitat categories, employing a Poisson error distribution. In addition, we performed ensemble-specific and species-specific GLMMs (for all species with a sample size ($n$) > 20, with abundance per species as the response variable). Model specifications were the same as for the species richness GLMMs, with sampling effort as an offset, habitat category and stratum as fixed effects (modelled as single-variable, additive and interaction effects), and location as a random effect. We checked all Poisson models for overdispersion and where present, corrected for it by including an individual-level random effect in the model [58].

### 2.4. Model Selection and Spatial Autocorrelation

To validate model assumptions, we plotted residual distributions, residuals *versus* fitted values and residuals *versus* each of the covariates [51]. We chose the final best-fit models by conducting a hierarchical model selection based on Akaike's Information Criterion corrected for small sample sizes (AICc). To quantify goodness-of-fit of each optimal model, we used marginal $R^2$ ($mR^2$) and conditional $R^2$ ($cR^2$; [59]). We performed independent Mantel tests (based on 1000 permutations) for each dataset (i.e., total abundance and total species richness) to test whether species composition could be explained by spatial autocorrelation. All Mantel test results were non-significant (see Supplementary Files, Tables S1 and S3), indicating that bat assemblage composition was uncorrelated with geographic distance and thus corrective measures were unnecessary. We considered the significance level as $\alpha < 0.05$, and all reported values are mean ± SE unless stated otherwise.

## 3. Results

### 3.1. Species Richness, Diversity and Dominance

We captured 3077 individuals of 46 species (1308 individuals of 42 species in CF and 1769 individuals of 40 species in forest fragments), with a total sampling effort of 16,356 *mnh* (Table 1). As overall sampling completeness was 93.9% (Jackknife 1), 93.7% for understory and 78.7% for all

upper forest strata, and the sample-based rarefaction curve reached an asymptote (Figure S2), our effort was deemed sufficient to adequately characterize the bat assemblages in our sampling sites [60].

**Table 1.** Number of individuals captured by family (bold), subfamily (bold and italic) and species (italic) in continuous forest, fragments, broken down by stratum. U: understory; LM: lower midstory; UM: upper midstory; SC: subcanopy.

| Taxa | Continuous Forest (CF) | | | | | Fragments (F) | | | | |
|---|---|---|---|---|---|---|---|---|---|---|
| | U | LM | UM | SC | CF Total | U | LM | UM | SC | F Total |
| **Mormoopidae** | | | | | | | | | | |
| *Pteronotus* cf. *rubiginosus* | 139 | 2 | | | **141** | 75 | 3 | 1 | | **79** |
| **Phyllostomidae** | | | | | | | | | | |
| ***Carolliinae*** | | | | | | | | | | |
| *Carollia brevicauda* | 25 | 4 | 1 | | **30** | 32 | 1 | 1 | | **34** |
| *Carollia castanea* | | | | | | 1 | | | | **1** |
| *Carollia perspicillata* | 330 | 22 | 14 | 6 | **372** | 709 | 51 | 46 | 9 | **815** |
| *Rhinophylla pumilio* | 132 | 11 | 8 | 2 | **153** | 220 | 23 | 42 | 8 | **293** |
| ***Stenodermatinae*** | | | | | | | | | | |
| *Ametrida centurio* | | 1 | 3 | 11 | **15** | | 2 | 3 | 1 | **6** |
| *Artibeus cinereus* | 12 | 4 | 9 | 4 | **28** | 13 | 2 | 12 | 1 | **28** |
| *Artibeus concolor* | 7 | 9 | 13 | 6 | **35** | 15 | 11 | 31 | 10 | **67** |
| *Artibeus gnomus* | 12 | 8 | 7 | 1 | **28** | 10 | 5 | 10 | 5 | **30** |
| *Artibeus lituratus* | 28 | 8 | 11 | 11 | **58** | 8 | 15 | 21 | 4 | **48** |
| *Artibeus obscurus* | 39 | 1 | 4 | 4 | **48** | 33 | 5 | 5 | 1 | **44** |
| *Artibeus planirostris* | 10 | 1 | 1 | 1 | **13** | 8 | 1 | 2 | | **11** |
| *Chiroderma trinitatum* | | | | 3 | **3** | | | | | |
| *Mesophylla macconnelli* | 23 | 3 | 12 | 6 | **44** | 6 | 1 | 3 | 2 | **12** |
| *Platyrrhinus* sp. | | | 2 | 1 | **3** | | | | 1 | **1** |
| *Sturnira tildae* | 1 | 2 | 1 | | **4** | 1 | 2 | 4 | 2 | **9** |
| *Uroderma bilobatum* | | | 1 | 1 | **2** | 4 | | 3 | | **7** |
| *Vampyriscus bidens* | 16 | 3 | 6 | 1 | **26** | 8 | 1 | 2 | 1 | **12** |
| *Vampyriscus brocki* | 1 | | 1 | | **2** | 3 | | | | **3** |
| *Vampyressa thyone* | | 1 | | | **1** | | | | | |
| ***Phyllostominae*** | | | | | | | | | | |
| *Chrotopterus auritus* | 4 | | | | **4** | 2 | | | | **2** |
| *Glyphonycteris daviesi* | 4 | | | | **4** | | | | | |
| *Glyphonycteris sylvestris* | 1 | 1 | | | **2** | | | | | |
| *Lampronycteris brachyotis* | | | | | | 1 | | | | **1** |
| *Lophostoma brasiliense* | 1 | | | | **1** | | | | | |
| *Lophostoma carrikeri* | 1 | | 1 | | **2** | 2 | | | | **2** |
| *Lophostoma schulzi* | 5 | | | | **5** | 4 | | | | **4** |
| *Lophostoma silvicolum* | 49 | | | | **49** | 17 | 1 | 1 | | **19** |
| *Micronycteris hirsuta* | | | | | | 1 | | | | **1** |
| *Micronycteris megalotis* | 2 | | | | **2** | 2 | | | | **2** |
| *Micronycteris microtis* | 5 | | | | **5** | 3 | | | | **3** |
| *Micronycteris sanborni* | | | | 2 | **2** | | | | | |
| *Micronycteris schmidtorum* | | | | | | 1 | | | | **1** |
| *Gardnerycteris crenulatum* | 22 | 1 | | | **23** | 26 | | | | **26** |
| *Phylloderma stenops* | 9 | | | | **9** | 7 | | | | **7** |
| *Phyllostomus discolor* | 3 | 6 | 7 | 1 | **17** | 3 | 38 | 42 | | **83** |
| *Phyllostomus elongatus* | 18 | | | | **18** | 6 | | | | **6** |
| *Phyllostomus hastatus* | 1 | | | | **1** | 1 | 1 | 2 | | **4** |
| *Tonatia saurophila* | 35 | 3 | 6 | | **44** | 32 | 4 | 3 | | **39** |
| *Trachops cirrhosus* | 70 | 1 | | | **71** | 29 | | | | **29** |
| *Trinycteris nicefori* | 4 | | | | **4** | 2 | 2 | 2 | | **6** |
| ***Glossophaginae*** | | | | | | | | | | |
| *Anoura caudifera* | 1 | | | | **1** | 1 | | 1 | | **2** |
| *Choeroniscus minor* | 1 | | 1 | | **2** | 6 | | | | **6** |
| *Glossophaga soricina* | 2 | | 1 | | **3** | 5 | | 1 | | **6** |
| *Hsunycteris thomasi* | 24 | | | | **24** | 16 | 1 | | | **17** |
| ***Desmodontinae*** | | | | | | | | | | |
| *Desmodus rotundus* | 8 | | | | **8** | 3 | | | | **3** |
| TOTAL | 1046 | 92 | 110 | 61 | **1308** | 1316 | 170 | 238 | 45 | **1769** |

We captured 12 species that were unique to the understory, including the frugivore *Carollia castanea*, eight gleaning insectivores (*Glyphonycteris daviesi*, *Lampronycteris brachyotis*, *Lophostoma*

*brasiliense*, *L. schulzi*, *Micronycteris hirsuta*, *M. megalotis*, *M. microtis*, and *M. schmidtorum*), the gleaning animalivores *Chrotopterus auritus* and *Phylloderma stenops*, and the hematophagous *Desmodus rotundus*. In contrast, five species were captured exclusively in the upper forest strata (see Table 1): the frugivores *Ametrida centurio*, *Chiroderma trinitatum*, *Platyrrhinus* sp., and *Vampyressa thyone*, as well as the gleaning insectivore *Micronycteris sanborni*. *Uroderma bilobatum* was captured solely in upper forest strata in CF, but within all forest strata in fragments.

Comparison of diversity measures between CF and fragments (Figure 1) revealed an overall decrease in species richness ($q = 0$) and both diversity indexes ($q = 1$ and $q = 2$) due to fragmentation. Species richness is more variable across strata within CF, tending towards higher richness in the understory for 1 ha and 10 ha fragments when comparing each stratum separately. However, grouping all upper forest strata reveals higher species richness than in the understory across habitat categories, although overlapping confidence intervals indicate these differences not to be significant. For $q = 1$ and $q = 2$, we found a more defined separation across strata, although the increased diversity (here meaning the effective number of common and dominant species, respectively) is also evident for the ungrouped upper forest strata *versus* the understory. Overall, this indicates that both CF and upper forest strata have more species with similar relative abundances (i.e., higher evenness, lower dominance), while fragments and the understory are dominated by a few highly-abundant species (i.e., lower evenness, higher dominance).

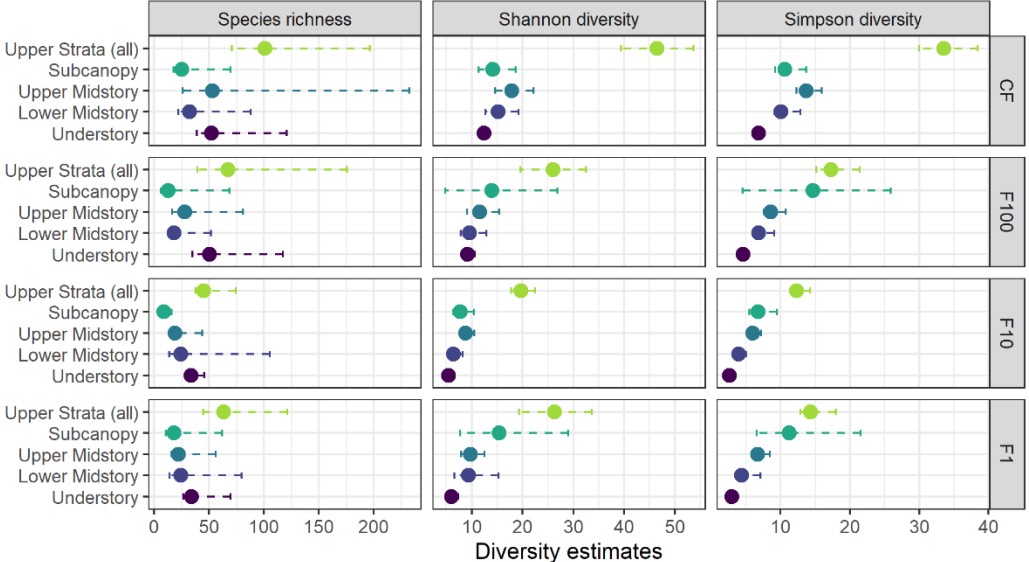

**Figure 1.** Species diversity estimates by habitat category and stratum, with 95% lower and upper confidence limits (dashed line). These estimates are produced by iNEXT and correspond to the Hill numbers $q = 0$ (species richness), $q = 1$ (Shannon diversity), and $q = 3$ (Simpson diversity). "Upper Strata (all)" refers to the pooled data of both midstory layers and subcanopy, for further comparisons with the understory. CF: continuous forest, F100: 100 ha fragments, F10: 10 ha fragments, F1: 1 ha fragments.

*3.2. Species Composition and Abundance*

To support the interpretation of diversity measures focusing on species richness, we compared sample-based rarefaction curves (Figure S2), and found differences between CF and fragments for the understory (Anderson-Darling k-sample test; $AD = 8.844$, $P = 0.002$) and the subcanopy stratum ($AD = 8.704$, $P = 0.002$). In general, the understory and lower midstory curves were steeper (i.e., dominated by a few common species), than those for the upper midstory and subcanopy.

Turnover and community change metrics (Figure 2) also suggest substantial fluctuations in the species present in CF *versus* fragments, differing by over 60% and more markedly for the upper strata (midstory and subcanopy layers) and for 1 ha fragments (Figure 2A). The mean rank shift (Figure 2B)

indicates considerable reshuffling of species from CF to 100 ha fragments, particularly for lower midstory and subcanopy. This reordering is less pronounced in the understory and upper midstory (with subcanopy presenting the largest shift).

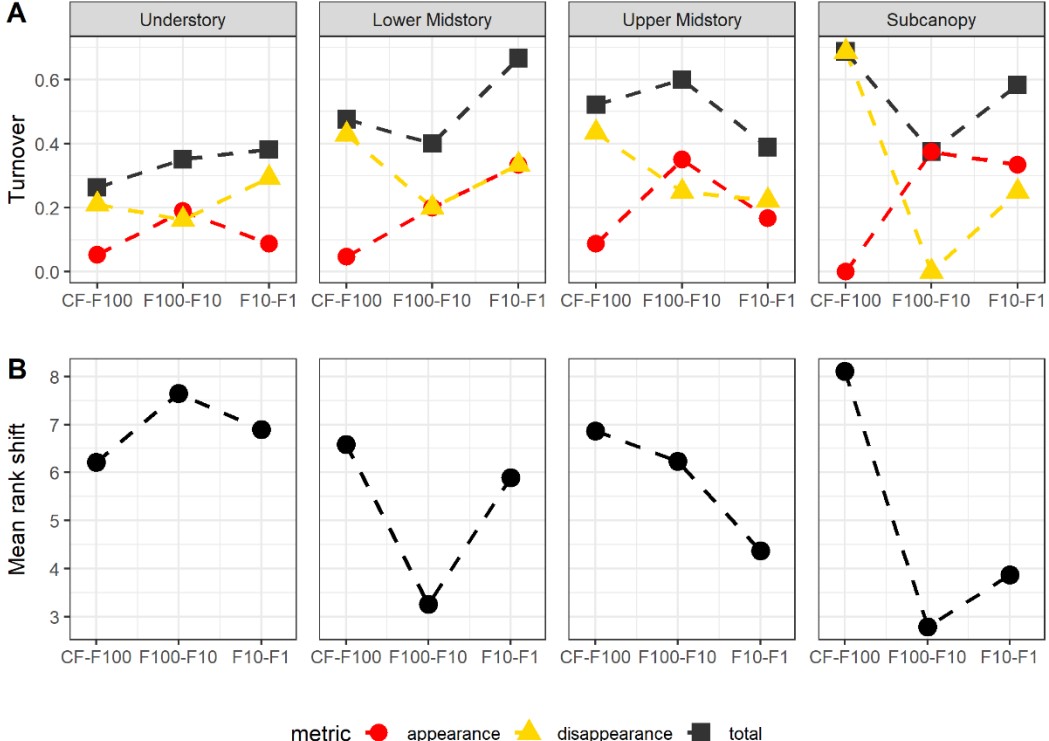

**Figure 2.** (**A**) Turnover, and (**B**) mean rank shifts for bat assemblages in continuous forest (CF), and in 100 ha, 10 ha and 1 ha forest fragments. Turnover is shown as total (black, square), appearances (red, triangle), and disappearances (yellow, triangle). CF-F100 denotes the shift from CF to 100 ha fragments, F100-F10 is the shift from 100 ha to 10 ha fragments, while F10-F1 is the shift from 10 ha to 1 ha fragments.

### 3.3. Species-Specific Strata Associations

*Artibeus cinereus* (Acin), *A. concolor* (Acon), *A. gnomus* (Agno) and *A. lituratus* (Alit; all four canopy frugivores) were significantly and positively associated with both midstory and subcanopy layers, while being significantly and negatively associated with the understory layer (Figure 3). *Phyllostomus discolor* (Pdis, gleaning omnivore) and *V. bidens* (Vbid, gleaning canopy frugivore) were positively associated with the midstory layers, while also being significantly and negatively associated with the understory layer. *Mesophylla macconnelli* (Mmac) was only associated positively for upper midstory and subcanopy, while *R. pumilio* (Rpum) was only associated with the lower midstory (see Table S6 for modelling results). In contrast, *C. perspicillata* (Cper) was significantly and negatively associated with the subcanopy and *Pteronotus* cf. *rubiginosus* (Prub) was significantly and negatively associated with the upper midstory.

Only the frugivores *A. lituratus* and *R. pumilio* presented a significant interaction effect between stratum and habitat category (Figure 4, Table S5). Both *R. pumilio* and *A. lituratus* had higher capture rates in the midstory layers of forest fragments than in CF. However, while the capture rate of *R. pumilio* was also higher in the subcanopy of forest fragments than in CF, for *A. lituratus*, the capture rate in the sub-canopy was higher in CF than in fragments.

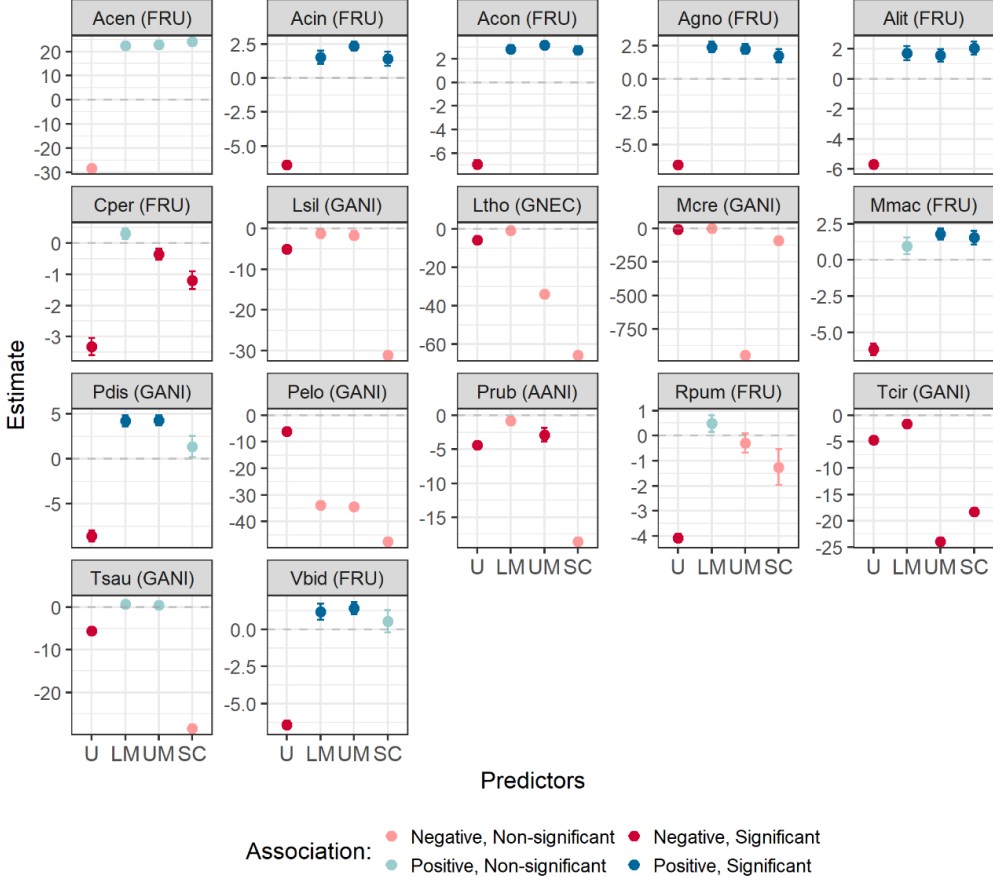

**Figure 3.** Parameter estimates (mean ± SE) of species-specific GLMMs, with stratum as a fixed effect, for all species with a sample size (n) > 20. Blue dots represent a positive estimate, while red dots represent a negative estimate. Forest stratum abbreviations: U = understory; LM = lower midstory; UM = upper midstory; and SC = subcanopy. Three species with sample size (n) > 20 are not represented (*A. obscurus*, *A. planirostris* and *Carollia brevicauda*), as their top model was the null model.

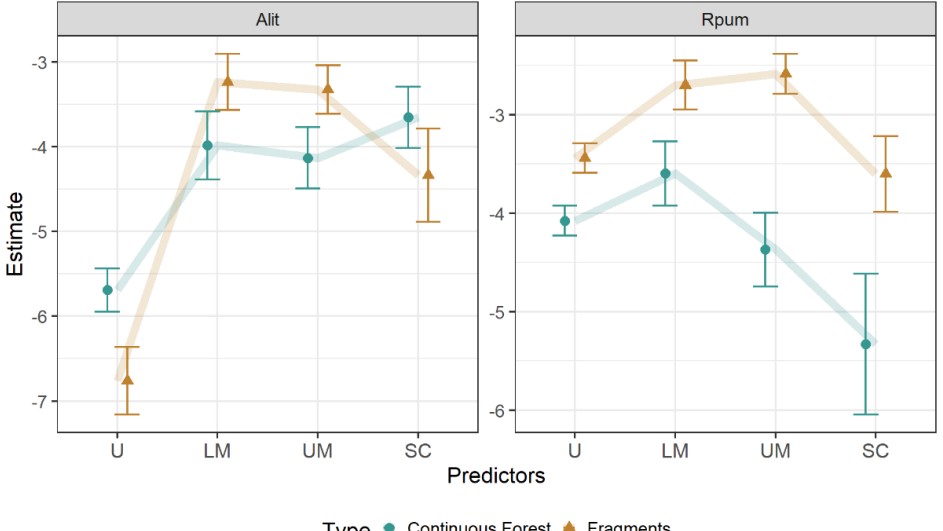

**Figure 4.** Parameter estimates (mean ± SE) of species-specific GLMMs, for interaction effect between stratum and habitat category, for the two species with stratum:habitat as the top model: *Artibeus lituratus* (Alit) and *Rhinophylla pumilio* (Rpum).

*3.4. Modelling Fragmentation Effects*

Model selection showed that stratum was a significant predictor of both richness and abundance (Tables S1 and S3). For assemblage species richness (Table S1), there was no significant interaction between stratum and habitat. Lower and upper midstory, as well as subcanopy, all exhibited higher species richness than the understory (Table S2). However, with standardized abundance, the best model revealed a significant interaction between stratum and habitat (CF *versus* fragments), although we could not disentangle the effect of fragment size: both upper and lower midstory layers had significantly higher bat abundance in fragments, while the subcanopy had significantly higher abundance in CF sites (Table S4).

At the ensemble level, only the abundance of gleaning animalivorous and frugivorous bats exhibited a significant interaction between stratum and habitat (Table S3). For frugivores, we found a similar pattern to that of the total abundance models, although dependent on fragment size, with higher frugivore abundance in the upper midstory of 1 ha fragments, and lower abundance in the subcanopy of 10 ha fragments, relative to CF sites. For gleaning animalivores, fragments had the same pattern (higher abundance levels in understory, lower and upper midstory, with a significant drop in the sub-canopy layer), while CF sites maintained similar bat abundance throughout all four layers. Two guilds (*i.e.,* aerial insectivores, sanguinivores) were represented by too few captures, species, or both to warrant model selection analyses. However, 214 out of 220 *P.* cf. *rubiginosus* (aerial insectivore) individuals were captured in the understory, while all *D. rotundus* (sanguinivore) individuals were captured in the understory ($n = 11$).

## 4. Discussion

A wealth of studies have examined the effects of habitat fragmentation on tropical understory bat assemblages [23]. However, studies surveying bats in the upper forest layers are rare and those contrasting assemblage patterns across strata in both CF and forest fragments are even rarer. By simultaneously sampling in the understory, midstory and subcanopy of CF and forest fragments we show that fragmentation modulates the vertical stratification of bat assemblages in the BDFFP landscape, leading to a substantial reduction of bat diversity in the upper forest layers in smaller fragments (< 10 ha) relative to CF and 100 ha fragments.

*4.1. Vertical Stratification in CF and Forest Fragments*

Vertical stratification was evident in both CF and forest fragments, independently of fragment size. These results align with previous findings of marked species structuring along the vertical axis in tropical America (e.g., [33,61]), Africa (e.g., [17]) and Asia (e.g., [15]) and thus provide additional evidence that vertical stratification is a key structuring feature of tropical bat assemblages.

Although the understory bat assemblages had higher species richness estimates than each of the upper forest strata in isolation, this pattern was reversed when data from both midstory layers and the subcanopy were pooled (upper strata; Figure 1). The difference in the understory and upper strata was particularly evident once the influence of rare and dominant species was reduced (i.e., Hill numbers $q = 1$ and $q = 2$), leading to between one to two times more estimated species in the upper forest strata than the understory in both CF and fragmented sites. Through the study of stable isotopes, Rex et al. [34] confirmed that several species (such as *R. pumilio*, *P. elongatus* and *L. silvicolum*) are actually foraging in the canopy although their capture numbers are higher in the understory; in combination with our results, this suggests that in both CF and forest fragments the upper forest strata might offer a higher diversity and abundance of food resources, which are likely to be explored by a more diverse pool of phyllostomid species.

### 4.2. Species-Specific Strata Associations

In general, across the BDFFP, the understory is dominated by a few common species, such as the frugivores *Carollia perspicillata* and *Rhinophylla pumilio*, and the aerial insectivore *Pteronotus* cf. *rubiginosus* [62,63]. These three species also occur in the upper forest strata, and both frugivores appear to utilize more forest strata in fragments, particularly 1 ha and 10 ha sites. This could indicate lower resource availability in the understory of smaller forest fragments, and lead individuals to occupy and forage within all forest strata. However, the vegetation density of the understory layer is higher post-fragmentation [64], which could also indicate these three species are flying higher to avoid clutter. In addition, bat activity is also linked to a number of factors beyond resource dynamics and abundance —such as weather, predation risk, roost availability, and reproductive stage [65–69]— which could influence their vertical movement patterns in fragmented or disturbed sites. Castro-Arellano et al. [70] found that logging had a greater effect on frugivores that foraged only in the understory than species that foraged in multiple forest strata. A successful response to fragmentation could be vertical plasticity, even with species that utilize only the understory in continuous primary forests.

In line with other studies [16,30,70], the upper forest strata were dominated by stenodermatines, while species from the Carolliinae subfamily were associated with the understory. The latter specialize in the fruits of understory plants, such as *Piper* and *Vismia* sp., whereas stenodermatines are known to forage across various forest strata [33]. By acting as seed dispersers of plants of both understory and upper strata, both subfamilies likely complement each other in enhancing second growth successional processes.

The relative abundance of several species shifted across strata and across habitat type; for example, although *P. discolor* appears to be using all forest layers within CF sites, it is almost exclusively found in the midstory layers of 100 ha and 10 ha fragments, and completely absent from the 1 ha fragments. *P. discolor* is mostly a canopy forager [16,30,71] and this suggests that the species may initially respond to fragmentation by exploring different strata, before disappearing from the smaller fragments. Other species that exhibited a similar pattern are the frugivores *Artibeus obscurus*, *A. gnomus*, *A. cinereus* and *Mesophylla macconnelli*, as well as the insectivorous *Tonatia saurophila*; all of these species (including the omnivore *P. discolor*) are adapted to highly-cluttered spaces [30], a characteristic associated with higher than average extinction risk [62,72,73].

Patterns of diversity and abundance can reflect different ecological conditions [74]. Top predators (e.g., *Chrotopterus auritus*; [75]), are intrinsically rare, but generalist species that require extensive foraging areas can also have low population densities, as is the case of *Phyllostomus hastatus* [76,77]. We only captured five *P. hastatus*, but when we accounted for effort, they were 11 times more likely to be captured within the upper forest strata of fragments than the understory of CF sites. This was also the case for the gleaning insectivore *Trinycteris nicefori*. This can be explained by food availability, as resources for carnivorous and insectivorous species are more abundant and diverse in intermediately-disturbed areas [78]. On the other hand, our only vampire bat species (*D. rotundus*) was captured exclusively in the understory which is likely a reflection of their preferential diet of non-arboreal mammals as has been reported in other studies [16,30,79,80]. However, and accounting for effort, *D. rotundus* was three times less likely to be captured in the fragments than CF (and 9 times less likely in 1 ha fragment sites), probably a consequence of low mammalian prey availability in the understory of the BDFFP forest fragments.

### 4.3. Effects of Fragmentation on the Vertical Stratification Structure

In general, bat assemblages in the upper forest strata were more diverse and stable in response to fragmentation than those associated with the understory layer. However, species turnover and rank shifts were more pronounced in the subcanopy, even in 100 ha fragments, indicating some degree of loss in the vertical structure of bat assemblages of the BDFFP landscape.

By combining canopy variables collected with portable canopy profiling lidar and airborne laser scanning surveys with long-term forest inventories, Almeida et al. [64] showed that even in the larger

BDFFP fragments, canopy height was reduced up to 40 m from the edge. This study further showed that near fragment edges, the density of understory vegetation was higher and midstory vegetation lower, reflecting the reorganization of the canopy as a result of increased regeneration of pioneer trees (mostly *Vismia* and *Cecropia* sp.) following post-fragmentation mortality of large trees. These changes in the physical structure of the forest layers of the BDFFP forest fragments, which are likely associated with changes in food availability, are probably the main driver of the changes observed in the vertical stratification of the bats inhabiting this landscape (e.g., *A. lituratus* feeds on mass-fruiting trees such as *Ficus* sp. that tend to dominate forest canopies [81]). Higher mortality of large canopy trees in forest fragments than in CF [82] might explain the observed lower capture rate of *A. lituratus* in the subcanopy of forest fragments than relative to the subcanopy of CF).

It is important to note that in the BDFFP deforestation was episodic and not continuous and that the fragments are embedded within a matrix of advanced second-growth vegetation, which may result in greater availability of food resources within the BDFFP compared to other fragmented landscapes. Additionally, the area is not affected by other anthropogenic threats that can alter forest structure such as forest fires or selective logging [7]. In "real-world" landscapes, changes in the three-dimensional structure of forest fragments are likely to be more severe, potentially translating into more conspicuous changes in the vertical stratification of bat assemblages.

## 5. Conclusions

Although bats are known to use the whole range of forest strata, research devoted to the impacts of forest fragmentation in tropical bats has until recently been mostly limited to the understory layer. By investigating patterns of diversity and assemblage structure resulting from both vertical stratification and fragmentation, our study adds to the understanding of the effects of habitat modification on Neotropical bats and will hopefully aid in the development of more effective conservation plans. Our results suggest that the maintenance of complex vertical vegetation structure is key for the conservation of Neotropical bats in human-modified landscapes and as such habitat restoration plans in fragmented landscapes should strive to enhance the multidimensionality of secondary forests.

**Supplementary Materials:** The following are available online at http://www.mdpi.com/1424-2818/12/2/67/s1, Figure S1: Map of the BDFFP experimental area in the Brazilian Amazon, Figure S2: Species-richness curves for the full dataset, and for understory-level and upper strata-level, Figure S3: Rank-abundance curves by habitat category and strata, Table S1: Top three GLMMs for predicting species richness (total dataset and by guild), Table S2: Parameter estimates for the species richness GLMMs, Table S3: Top three GLMMs for predicting bat abundance (total dataset and by guild), Table S4: Parameter estimates for the bat abundance GLMMs, Table S5: Top three species-specific GLMMs for predicting bat abundance, Table S6: Parameter estimates for the species-specific GLMMs predicting bat abundance.

**Author Contributions:** Conceptualization, I.S., R.R. and C.F.J.M.; Data curation, I.S., R.R., A.L.-B., F.Z.F. and C.F.J.M.; Formal analysis, I.S.; Funding acquisition, R.R., A.L.-B. and C.F.J.M.; Investigation, I.S., R.R., A.L.-B., F.Z.F. and C.F.J.M.; Methodology, I.S., R.R. and C.F.J.M.; Project administration, C.F.J.M.; Supervision, R.R. and C.F.J.M.; Validation, I.S., R.R., A.L.-B., F.Z.F. and C.F.J.M.; Visualization, I.S.; Writing – original draft, I.S. and R.R.; Writing – review & editing, I.S., R.R., A.L.-B., F.Z.F. and C.F.J.M. All authors have read and agreed to the published version of the manuscript.

**Funding:** Funding was provided by the Portuguese Foundation for Science and Technology to C.F.J.M. (PTDC/BIA-BIC/111184/2009), R.R. (SFRH/BD/80488/2011) and A.L.-B. (PD/BD/52597/2014). F.Z.F. was supported by a fellowship from Coordenação de Aperfeiçoamento de Pessoal de Nível Superior (CAPES). Additional funding was provided by Bat Conservation International student research fellowships to A.L.-B and R.R. and by ARDITI – Madeira's Regional Agency for the Development of Research, Technology and Innovation (grant M1420-09-5369-FSE-000002) to R.R.

**Acknowledgments:** We would like to thank the volunteers and field assistants that participated in data collection as well as the BDFFP for logistical support. This research was conducted under ICMBio (Instituto Chico Mendes de Conservação da Biodiversidade) permit (26877-2) and constitutes publication number 782 in the BDFFP technical series.

**Conflicts of Interest:** The authors declare no conflict of interest.

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
