# Peer review of "Effects of Forest Fragmentation on the Vertical Stratification of Neotropical Bats"

_diversity, doi:10.3390/d12020067_

Round 1

Reviewer 1 Report

This paper compares the vertical stratification of bat assemblages between continuous and fragmented forests in Amazonian Brazil.  The poorly sampled canopy was found to be more diverse and abundant than at ground level.  Fragmented forest also had an affect on stratification compared to continuous forest.  Overall, it is a well written study.

Not crucial, but from a different region in South America and contemporaneous with some of the earlier publications on bat vertical stratification is the paper “Lim, B.K., and M.D. Engstrom. 2001. Bat community structure at Iwokrama Forest, Guyana. Journal of Tropical Ecology, 17: 647-665.”

My only real comment is the lack of updated taxonomy or at least discussion of why names weren’t used.  For example, Pteronotus cf. rubiginosus could also be the newly described species Pteronotus alitonus (Pavan et al. 2018).  Is Artibeus bogotensis not recognized from the Manaus region?  In the Guianas it is usually found in forested areas instead of Artibeus cinereus, which is more associated with savanna.  Artibeus gnomus is the most common forest species of these 3 (Lim et al. 2008).  Platyrrhinus helleri is now restricted to Central America and northwestern South America, and the Guianas may have at least 3 sympatric species (P. incarum, P. fusciventris, and P. guianensis) based on Velazco and Lim (2014).  The genus should be Vampyriscus for both bidens and brocki, which is a more straightforward taxonomic fix.  And Vampyressa pusilla should be V. thyoneMicronycteris microtis should be synonymized under M. megalotis as suggested by Porter et al. (2007).  Unfortunately, they did not make a formal taxonomic change, but their molecular study did not support the separation of 2 taxa.  Mimon crenulatum should be Gardnerycteris crenulatum and Lonchophylla thomasi should be Hsunycteris thomasi.

Reviewer 2 Report

General comments:

This study evaluates the role that forest fragmentation has on vertical stratification of Neotropical bat assemblages. Despite the knowledge that fragmentation has complex multidimensional effects on tropical fauna, this topic remains poorly studied and the current manuscript has the potential to provide valuable context for management. I found the manuscript to be well written and comprehensive in the suite of analyses presented. I have no major concerns, however I do have a number of minor comments which need to be addressed. They primarily focus on the description and interpretation of the results, which I found to be the least polished aspect of the manuscript. I found some aspects that appear to contradict figures and the citation of tables and supplementary material to be incomplete or confusing due to a lack of correspondence. I feel these are all easily remedied with careful attention.

Minor comments:

Ln 168 I did not find a discussion of the hierarchical partitioning results anywhere. If they are not discussed than they should not be included in the methods or results.

Ln 233 Based on Figure3 Acen does not appear to be significantly associated with any strata.

Ln 233 Acon Agno and Alit were all negatively associated with understory. May consider revising this sentence to include this information as it reinforces the idea that they are canopy frugivores.

Ln 248 I think this should refer to File S3 rather than S2.

Ln 253 Files S1, S2 and S3 all had modelling results referred to in this section

Ln 253 File S1, Table 2

Ln 255 are these estimates in a table somewhere? I did not see them.

Ln 256 There is no table 3 in any of the supplementary files. What does standardized abundance refer to?

Ln260 It appears that Gleaning Animalivores also had a significant interaction based on File S2 Table 2

Ln 339 I think the statement regarding Desmodus rotundus is too strong and evidence indicates that D. rotundus can persist and achieve high relative abundance in highly fragmented areas or pasture areas. If this is meant to be specific to the study area please indicate that, otherwise I suggest the authors revise or delete the comment.

See for example:

Bolívar‐Cimé, B., Cuxim‐Koyoc, A., Reyes‐Novelo, E., Morales‐Malacara, J.B., Laborde, J. and Flores‐Peredo, R. (2018), Habitat fragmentation and the prevalence of parasites (Diptera, Streblidae) on three Phyllostomid bat species. Biotropica, 50: 90-97. doi:10.1111/btp.12489

File S1 Table1 Please include description in the table legend or file description of what GANI, GFRU, GNE refer to as well as Mantel r and what the colors signify.

Reviewer 3 Report

Review of the manuscript “Effects of forest fragmentation on the vertical stratification of Neotropical bats” by Silva, I et al

This paper deals with the interaction of vertical stratification on rainforests with fragmentation. It is a very interesting topic and there is an urgent need to understand the impact of these phenomena on biodiversity. Vertical stratification is one of the most important and overlooked drivers of the remarkably high biodiversity in tropical forests whereas, fragmentation is one of the most prevalent threats to these areas. Moreover, the BDFFP project is one of the most important to understand the role and impacts of forest framentation across biological groups in the tropical regions. Thus, the topic is very interesting and the publication of these analyses should be encouraged.

On a overall analysis of the manuscript is well written and structured. However, I do have a few concerns and questions about the manuscript.

Main concern:

The vertical strata grouping where the authors contrast the understorey with all the upper strata (low midstorey, upper midstorey and subcanopy. Because there is no interval between the height strata the authors need to justify why they chose this grouping of strata (understorey versus all upper strata) and not some of the alternatives such as comparing the two lower strata with the two higher strata. This is my main concern because as the authors point – correctly in lines 288-289 - in the discussion other strata groupings can yield different results. Was it because of the most of the published literature analyses bat captures only up to 3 meters (height of the ground nets?)

Abstract:

The very interesting result of the A. lituratus and Rhinophylla pumilio shifting their vertical stratification pattern and strata association between CF and fragments should be further explained by including what where the changes for these species (lines 33-34). This will clarify the results part of the abstract.

Key words: Ok, but maybe you could spare Neotropics

Introduction:

Good introduction describing the patterns of bat response to fragmentation from the literature:

gleaning animalivorous are more susceptible than frugivores or nectarivores and canopy foragers less sensitive than understorey species

It is very good to have a clear set of questions to be addressed with acompanying predictions for each.

Methods:

One concern is that the separation of fragments from continuous forest by less than 100m in a few sampling sites could be a confounding variable and have a significant impact on the results. So, it is needed information to answer the question “Does the separation of 80-650m meters from fragments to continuous forests warrants an isolation effect for some of the bat species analysed, and thus, an effect of the fragmentation? “

The authors already refer that to maintain fragment isolation a strip of 100m wide is (lines 108-109). Thus, you should provide details also on when were the strip clearances before the start of the field work, before the one that took place in 2014.

Overall the statistical analysis was sound and solid using appropriate and recent tools and methods. The authors used the available methods’ options to correct for sampling differences (offset), potential spatial correlation (locations as random factor). Models were correctly specified and in the Poisson models there was the necessary care with overdispersion, by adding an individidual ramdom factor when needed.

Results:

line 217 please clarify which of the upper strata you are referring to (subcanopy?)

line 233 From the figure 3, I think that it is the species Acin that has a positive significant association with all the upper strata, and not Acen. If that is the case, the abbreviation in the text should be corrected.

line 239-240 the authors refer to Pteronotus cf. rubiginosus (Prub) but I that species is missing from Figure 3 .. or is it a typo in the graph Ppar that should be Prub?

Line 248-249 please rewrite this sentence to imporve its clarity.

Discussion:

lines 337-339. There is mounting evidence that contradicts your conclusion that Desmondus rotundus cannot persist in highly fragmented areas. But I believe that the effect of fragmentation that you found is correct. The problem arises when the authors extend the pattern they observed of the impact of fragmentation on D. rotundus to “all” highly-fragmented areas. This species can thrive in very disturbed and fragmented areas when they have alternative prey such as livestock. Please re-write this interesting result and your conclusion.

Section 4.3 Effects of fragmentation on the vertical structure.

In my opinion the authors should include a brief discussion of the Aribeus lituratus results, because it is a frugivorous species associated with the upper strata, but that also showed a pattern consistent with a negative impact of fragmentation. This is evidence that contradicts the general pattern reported in the literature that frugivores associated with upper forest strata are less sensitive to fragmentation (lines 64-65 and 71-72 in the introduction)

References:

I found several inconsistencies on several references: they have the journal name without capital letter in the second word.

The first examples include (line 397 Journal of biogeography; line 460 Forest ecology and management), but there are several more.
